# Graph-based Algorithms for Nearest Neighbor Search with Multiple Filters

## Abstract

We study nearest neighbor search with filter constraints (MultiFilterANN): given a query vector with a discrete set of labels $S$, retrieve the (approximately) closest vector from a dataset under the constraint that $S$ must be a subset of the labels of the retrieved vector. There has been a burgeoning interest in this problem on the practical side, due to its strong motivation from search and recommendation applications where vector labels correspond to real world attributes such as date, price, or color. On the theoretical side, this problem generalizes the subset query problem, which asks us to only determine if $S$ is a subset of some set in the dataset, without retrieving the closest vector.

In this work, we present a systematic study of MultiFilterANN,. Theoretically, we demonstrate the power of graph-based algorithms in two ways:

- We design provable algorithms with the best known space-time tradeoffs for MultiFilterANN in the large filter regime by carefully incorporating ANN algorithms into known subset query algorithms.
- We demonstrate lower bounds for popular algorithms for MultiFilterANN, showing that they can catastrophically fail even on simple data/label sets.

Our theoretical results inspire our empirical approach, where we extend practical graph indices for standard nearest neighbor search to MultiFilterANN by augmenting the (greedy) search procedure with a penalized distance function that captures filter constraints. Our empirical algorithm is competitive with existing state of the art solutions which are tailored for one or two filters, while also seamlessly generalizing to any number of filters without any modifications. Lastly we release multiple novel datasets for MultiFilterANN, filling in a noticeable gap in literature.

## 1 Introduction

Driven by advances in machine learning, the capabilities of embeddings to faithfully represent semantic relationships between real world objects such as video, images, and text via geometric distances has dramatically increased. This has prompted the study of new algorithmic questions centered around the analysis of large scale vector data. A prominent example is the nearest neighbor problem with filter constraints (MultiFilterANN): we are given a dataset of $n$ high-dimensional vectors $X$ where every vector $v \in X$ has a set of associated labels $S_v \subseteq [m]$[1]. We wish to design a data structure, which given a query vector $q$ with its own set of labels $S_q \subseteq [m]$ and a target $k \in \mathbf{N}$, outputs the $k$ (approximately) closest vectors to $q$ in $X$, subject to the constraint that $S_q$ *must* be a subset of the label set $S_v$ of each of the vectors $v$ returned.

The problem is well motivated by practical constraints in search and recommendation. It captures the realistic scenario of combining vector search with keyword matches. For example in image search, we might only be interested in images shot in Paris. In product recommendations, a shopper may only wish to view items with certain specifications of size, make, and model. Likewise for text data, we may only want documents that contain a set of given keywords to be retrieved. In general, hybrid keyword semantic search is a fundamental task in many modern systems. Leading vector search companies such as Weaviate Weaviate (2024) and Pinecone Pinecone (2024) offer "hybrid search" as a service, with the motivating example being a combination of hard matches based on keywords

---

[1]$[m] := \{1, \ldots, m\}$

and a vector distance based on the dense LLM-based embeddings. Our formulation models the case where the returned data points must satisfy *all* the labels of the query.

Beyond its practical appeal, the problem is also rich theoretically: it generalizes the classic subset query problem from the algorithms literature, where we wish to quickly determine if a given set $S_q \subset [m]$ is a subset of a collection of $n$ subsets of $[n]$ (i.e., MultiFilterANN without vector search).

Many recent empirical solutions have been proposed for MultiFilterANN, with most work focusing on the single-filter regime and more recently, $|S_q| = 2$. These works demonstrate that many existing algorithms for the vanilla nearest neighbor search problem can be extended to handle the case of few filters. Nevertheless, while tremendous practical progress has been made, substantial gaps remain in our understanding of MultiFilterANN, captured by the following questions:

- On the empirical side, can we design algorithms that scale as the numbers of constraints, i.e. $m$ and $|S_q|$, grow? What are the limits of current approaches deployed for the small-filter case?

- Likewise, what can we say theoretically for MultiFilterANN? First, do the existing practical algorithms for the small-filter case have provable guarantees? And secondly, can we obtain algorithms for MultiFilterANN whose performance is comparable to the best known algorithms for the subset query problem? Note that if $m$ is constant, there is a simple naive algorithm with a $2^m$ factor space overhead which constructs a nearest neighbor data structure corresponding to every subset of $[m]$. Thus, the problem is especially interesting when $m$ is large.

**Our Contributions.** We initiate a broad and systematic study of the MultiFilterANN problem.

- On the theoretical side, we give an algorithm with $o(2^m)$ space overhead and $o(n)$ query time (in certain regimes), matching some of the best known space and query time trade-offs for subset query problem, while additionally handling the nearest neighbor search component. Our algorithm is flexible and capable of incorporating graph-based indices.

- Bridging theory and practice, we construct simple datasets demonstrating failure modes of popular solutions for the small-filter regime that are based solely on clustering.

- Inspired by our theoretical understanding, we give a practical algorithm for MultiFilterANN based on DiskANN, a popular graph-based method for approximate nearest neighbor search. Our method easily generalizes to *any* number of filters, and comes with theoretical guarantees under assumptions on bounded doubling dimension of the data. It can return a specified number of near neighbors (all satisfying the label constraints), and we measure accuracy using Recall@k.

  At a high level, we change the greedy search procedure of the query phase of DiskANN, carefully incorporating both vector distance and label information in an *asymmetric* manner (see Section 5). Crucially, a query search can utilize all edges of the graph index, even those that connect vertices which don't satisfy the filter constraint of the query. In summary, our algorithm improves recall by up to $40\%$ for comparable latency over prior baselines.

- Lastly, we also release a novel vector dataset with multi-filter constraints and describe new synthetic datasets, filling a noticeable gap in the literature. While the MultiFilterANN problem is demonstrably important in practice, publicly available datasets which capture the complexity of the problem are few and far between. For example, existing datasets (such as those in Wang et al. (2022)) can either be easily reduced to the one filter case, since the total number of combinations is very small, or can be brute forced since there are very few data points per label. Our dataset is quite natural: the vector consist of embeddings of Wikipedia pages and filters correspond to common keywords. We hope our new dataset becomes a benchmark for future work on MultifilterANN; see the anonymous link `https://www.dropbox.com/scl/fi/h5om8cwxosrqhc9ortk19/filterann-dataset-link.txt?rlkey=syjlvzcxvr59m4ule3uv9pyhv&st=0gtcgfod&dl=0`.

## 2 PRELIMINARIES

Our 'base' dataset $X$ consists of $n$ points in $\mathbb{R}^d$. In addition, every point $x \in X$ also has a set of labels (or filters) $S_x \subseteq [m]$. In this setting, $X$ is also referred to as a labeled dataset. A query $q \in \mathbb{R}^d$ also has a set of labels $S_q \subseteq [m]$. We define the Multi-FilterANN problem as follows.

**Definition 2.1** (Multi-FilterANN). *We are given a labeled dataset $X$ as above and a parameter $C \geq 1$. We want to create a data structure which on any query $q$, returns a point $x \in X$ guaranteeing that $S_q \subseteq S_x$ and $\|x - q\|_2 \leq C \cdot \min_{x' \in X, S_q \subseteq S_{x'}} \|x' - q\|_2$.*

This definition is the analogue of the theoretical work on nearest neighbor search Andoni et al. (2018). Thus we think of datapoints as having a vector component as well as a label component. We always $\ell_2$ (Euclidean metric) to measure vector distance. In our experiments, we instead report the standard Recall@k score, which is analogous to the empirical works on standard nearest neighbor search without filters. It denotes the fraction of the returned top-$k$ points that are among the true $k$ nearest neighbors of a query $q$. Note that while we allow for an approximation in the vector distance, we always require the subset condition to be satisfied.

The Multi-FilterANN problem is a strict generalization of the SubsetQuery problem, whose study goes back at least two decades Charikar et al. (2002).

**Definition 2.2** (SubsetQuery Problem). *We are given a collection $X$ of $n$ subsets of $[m]$. When a query $q \subseteq [m]$ arrives, we must determine if there exists an $x \in X$ such that $q \subseteq x$.*

Usually the SubsetQuery problem asks if there exists an $x$ such that $x \subseteq q$, but the formulation we state is equivalent by taking complements.

Using the language of prior works Gollapudi et al. (2023), in Multi-FilterANN we are solving the 'AND' case, where *all* the labels of the query must be present in the data points returned. This is as opposed to the much easier 'OR' case, where only one label of the query can be present. The 'OR' case can be reduced to multiple invocations of an algorithm for the single-filter case.

Lastly, we introduce the doubling dimension of a dataset. It is a well-studied measure of intrinsic dataset dimensionality in the context of nearest neighbor search in theory Gupta et al. (2003); Krauthgamer & Lee (2004); Indyk & Naor (2007); Har-Peled & Kumar (2013). Furthermore in practice, the performance of popular empirical algorithms can be analyzed via the doubling dimension Narayanan et al. (2021); Indyk & Xu (2023), and many real world dataset exhibit small intrinsic dimensionality Aumüller & Ceccarello (2019).

We define it for the vector component of a dataset. For any $x \in X$, radius $r > 0$, we use $B(p, r)$ to denote the ball of radius $r$ centered at $p$, i.e. $B(p, r) = \{q \in X : \|p - q\| \leq r\}$.

**Definition 2.3** (Doubling Dimension). *A vector data set $X$ has doubling dimension $\lambda$ if for any $p \in X$ and radius $r > 0$, $X \cap B(p, 2r)$ can be covered by at most $2^\lambda$ balls of radius $r$.*

## 3 RELATED WORK

Over the years, there has been extensive research on ANNS algorithms with the focus on enhancing recall, improving scale, cost-efficiency, distributed indexing, real-time index updates, and theoretical guarantees Bentley (1975); Beygelzimer et al. (2006); Muja & Lowe (2014); Bernhardsson (2018); Indyk & Motwani (1998b); Andoni & Indyk (2008); Zheng et al. (2020); Andoni & Razenshteyn (2015); Sundaram et al. (2013); Park et al. (2015); Liu et al. (2011); Jiang & Li (2015); Johnson et al. (2017); Baranchuk et al. (2018); Babenko & Lempitsky (2012); Ge et al. (2014); Arya & Mount (1993); Malkov & Yashunin (2016); Fu et al. (2019); Subramanya et al. (2019); Echihabi et al. (2019); Baranchuk et al. (2018); Sundaram et al. (2013); Chen et al. (2021); Xu et al. (2023); Guo et al. (2019). The interested reader may refer to the ANN benchmark efforts Simhadri et al. (2023); Aumüller et al. (2023). On the other hand, we have only recently begun scratching the surface when it comes to MultiFilterANN. Analytic DB-VWei et al. (2020) and VBase Zhang et al. (2023) present elegant solutions to integrate filtered ANNS queries into database systems, but their general-purpose nature means that these methods typically have higher latencies than some high-performance scenarios permit. CAPS Gupta et al. (2023) develops an algorithm for MultiFilterANN by combining data structures for ANN and subset query, but the performance seems to suffer significantly as the size of the label universe grows. SERF Zuo et al. (2024) makes clever modifications to the index construction phase of graph algorithms to enable them to support *single* range queries (e.g., filter by date). Importantly, these ideas do not generalize to multiple filters. More recently, ACORN Patel et al. (2024) and $IVF^2$ are also newer methods for filtered ANNS. The ideas in ACORN, however, do not scale well to even simple AND predicates. While $IVF^2$ is the state-of-art open source algorithm for MultiFilterANN, the algorithm is purely clustering based, and we demonstrate scenarios where

graph-based methods, if done right, can still outperform clustering based methods even in the MultiFilterANN problem.

The Subset Query algorithm also has several practical real-world applications, especially for keyword based search. In traditional search engines, each keyword maintains an inverted index of associated documents and the search essentially boils down to fast intersection methods of these inverted indices; see Wang & Suel (2019); Goodwin et al. (2017) and the citations therein for a more complete set of references. One can see our work as a natural step in the direction of fusing vector search with traditional keyword-based search ideas.

## 4 THEORETICAL ANALYSIS

We present an algorithm with strong theoretical (worst-case) guarantees for the Multi-FilterANN problem. We additionally take inspiration from our theoretical study to design our final empirical algorithm of Section 6. This is elaborated at the end of the section.

The starting point of our theoretical algorithm is to note that any Multi-FilterANN data structure must be able to solve the SubsetQuery problem. Thus, we first review a data structure from Charikar, Indyk and Panigrahy, denoted as CIP, for the SubsetQuery problem Charikar et al. (2002). The CIP data structure is among the best known when $m = O(\text{poly}(\log n))$. Note that as stated, CIP cannot handle the full generality of the Multi-FilterANN problem.

**Summary of their approach:**  The CIP data structure partitions the input dataset (of the Subset-Query problem) into multiple levels, thought of as a nested table. All levels, except the final one, hash a given query to only one entry of the table. This last level is a collection of disjoint sets $X = \cup_i X_i$ of the data points. The CIP data structure guarantees that (with high probability), for all $X_i$, either all datapoints in $X_i$ are valid responses to the query (i.e., the query is a subset of all the datapoints in $X_i$), or none are. However, the data structure needs to evaluate this for every group. Fortunately, Charikar et al. (2002) guarantees that we can construct the data structure so that only $o(n)$ groups are at the last level, so only $o(n)$ checks need to be made. A detailed summary of the CIP data structure and its formal guarantees are given in Appendix B.1 (see Theorem B.1).

**Adapting from SubsetQuery to Multi-FilterANN**  Our key idea is to replace each of the final groups in CIP with a nearest-neighbor data structure. If a group (in the final layer of the CIP data structure) is valid for the query, then we use a vanilla nearest-neighbor data structure to find a close vector to the query's vector. Note that this takes care of both the SubsetQuery requirement, as well as the vector search component, i.e., is applicable to the Multi-FilterANN problem. For theoretical guarantees, we may use Locality Sensitive Hashing Indyk & Motwani (1998a) in the final level of the CIP data structure or the DiskANN graph. It implies the following theorem, with a formal proof deferred to Appendix B.

**Theorem 4.1.** (Main theoretical guarantee for Multi-FilterANN) *There exists a data structure which, with probability* $0.99$, *returns a* $(1 + \varepsilon)-$*approximate filtered nearest neighbor on any query. This data structure uses* $\tilde{O}(n^{1-\delta}(m + dn^\delta + n^{3\delta})2^{O(m\sqrt{\delta}\log^2 m)})$ *space and on any query, performs at most* $O(n^{1-\delta})$ *set intersections and* $\tilde{O}(n^{1-\varepsilon\delta})$ *distance comparisons.*

The intuition of the above theorem is the difficulty of the MultiFilterANN problem is dominated by the subset query requirement. This is why the stated bounds of Theorem 4.1 generalize Theorem B.1.

Note that we can choose any unfiltered approximate nearest neighbor data structure at the lowest level of the CIP data structure, instead of LSH (see Corollary B.3). In particular, we can also use graph-based data structures, such as (the 'slow preprocessing version' of) DiskANN Indyk & Xu (2023) instead of CIP to obtain an identical guarantee as in Theorem 4.1, but using graph indices.

**Theorem 4.2.** (Graph theoretical guarantee for Multi-FilterANN) *There exists a data structure which, with probability* $0.99$, *returns a* $(1 + \varepsilon)-$*approximate filtered nearest neighbor on any query. This data structure uses* $\tilde{O}(n^{1-\delta}(m + n^\delta(1/\epsilon)^{O(\lambda)})2^{O(m\sqrt{\delta}\log^2 m)})$ *space and on any query, performs at most* $O(n^{1-\delta})$ *set intersections and* $\tilde{O}(n^{1-\delta}(1/\epsilon)^{O(\lambda)})$ *distance comparisons where* $\lambda$ *is the doubling dimension of the dataset.*

Furthermore, in practice graph-based indices are empirically among the state of the art for approximate nearest neighbor search Jayaram Subramanya et al. (2019), and our guarantees show that we can indeed invoke them for the more challenging Multi-FilterANN problem, without losing any theoretical power. We further justify the use of graphs for Multi-FilterANN in Section 6.

> **Lesson 1:** Graph-based nearest neighbor algorithms are powerful primitives for Multi-FilterANN.

Note that the theoretical algorithm we presented starts by routing queries to a small subset of *relevant* data (the hashing levels of CIP guarantee few groups at the lowest level). This allows us to perform a refined nearest neighbor search on only 'relevant' parts of the input dataset. This motivates a similar high-level actionable idea in any practical algorithm, where we route queries to the appropriate subset of the input data. This is discussed in our empirical algorithm design section, Section 6.

> **Lesson 2:** Query planning is important.

## 5 MOTIVATING OUR EMPIRICAL ALGO. BY STUDYING PRIOR FAILURES

We now additionally motivate our final empirical algorithm by presenting failure modes of prior approaches for the one or two filters regime. By demonstrating simple instances where such approaches catastrophically fail, we gain valuable insights towards our final algorithm.

**Parlay$IVF^2$ Counterexample.** We start with Parlay$IVF^2$, the winner of the recent ANN Benchmark challenge on nearest neighbor search with 2 filters (queries have at most 2 filters) Landrum et al. (2024). As a summary, the Parlay$IVF^2$ algorithm essentially constructs a clustering index for each label. When a query arrives, for each label in the query, it evaluates the distance of each cluster to the query, sorting the clusters in ascending order of distance. It keeps adding all the points from the nearest remaining cluster into a queue until the size of the queue is at least a pre-determined parameter. Then, it takes the intersection of these queues and evaluates the distances. In some simple instances, we can prove that in order to find the nearest point satisfying both the labels, Parlay$IVF^2$ must add almost all the points into the queue. The corresponding intersections of sets containing $O(n)$ elements can make Parlay$IVF^2$ take linear time if we want to obtain any non-zero recall.

**Lemma 5.1.** *There exists a size-$n$ one-dimensional dataset with two total labels such that the Parlay$IVF^2$ algorithm has query time $\Omega(n)$.*

This highlights that popular prior approaches for our problem, which use clustering based methods, are too 'local.' In the simple one dimensional aforementioned example, clustering based methods spend too much time focusing on the 'distance' part of the problem and ignore the subset match constraint. This forces them to iterate over many irrelevant points in the local vicinity of the query.

In contrast, that graph-based indices avoid this problem by having *long-range* edges, which in this instance, would allow us to quickly navigate to the end of the one dimensional dataset, without performing a linear scan through the number line. This prompts the following lesson.

> **Lesson 3:** Clustering indices alone are not enough for Multi-FilterANN. Long-range connections afforded by graph indices are needed.

We now discuss two prior graph-based approaches: FilterDiskANN Gollapudi et al. (2023) and NHQ Wang et al. (2022; 2024). While they are both graph-based, we give provable hard instances for these algorithms, motivating important modifications used for our final graph-based empirical algorithm.

**FilterDiskANN Counterexample.** The paper Gollapudi et al. (2023) was the first to empirically study nearest neighbor search with filter constraints, and they focused on the 1-filter regime (queries have 1 label only). At a high level, they construct a graph index (based on the vanilla DiskANN algorithm), and perform the following greedy search given a query $q$ with label $S_q$: traverse the graph only on edges with both endpoint vertices having label $S_q$. They start the greedy search procedure at a fixed starting vertex containing label $S_q$. Thus, for every label $\ell \in [m]$, they require the subgraph of all vertices with label $\ell$ to be connected in their graph index. While this approach works well for the $m = 1$ single-filter case, we show that it cannot generalize to large $m$.

In particular, if we require the subgraph corresponding to datapoints $X$ that are 'valid' for any query $q$ (the points in $X$ whose labels contain those of $q$) to be connected, then the graph must have $\Omega(n^2)$ edges. This holds for any graph with the aforementioned propriety (not just those based on DiskANN), making it prohibitive to store or initialize such graphs for large datasets.

**Lemma 5.2.** *There exist a labeled dataset $X$ of size $n$ with $m = O(\log n)$ total labels such that any graph index on $X$ with the property that the subgraph of points satisfying the label constraints of a query is connected, must have $\Omega(n^2)$ edges.*

Intuitively, the approach of Gollapudi et al. (2023) leads to a dense graph for large $m$ because queries are prohibited from traversing edges that are not directly relevant to the labels of the query. This restriction requires having many extraneous edges due to the combinatorial explosion of possible label sets of queries, leading to an almost trivial graph index.

---

**Lesson 4:** We must be allowed to use all edges of a graph index, even those that connect vertices not directly relevant to the query labels.

---

**NHQ does not handle subset queries.** Lastly, we study the NHQ algorithm Wang et al. (2022; 2024). It defines a "fusion distance" which is the geometric distance between the vector component of two points, multiplied by a term that depends on the symmetric difference of their label sets. The fusion distance for two points $x$ and $y$ is set to $\|x - y\|_2 \cdot \left(1 + \frac{\|S_x - S_y\|}{m}\right)$, where $\|S_x - S_y\|$ is the Hamming distance between the $0/1$ encodings of the sets $S_x$ and $S_y$. Using the fusion distance, they construct a graph index; when a query arrives, the fusion distance is used to perform greedy search on the graph. While this does work well for *exact* label set queries (i.e. if we wanted $S_x = S_y$ exactly), this approach can fail catastrophically for *subset* constraints.

**Lemma 5.3.** *There exists a one-dimensional dataset $X$ and a query $q$ such that an incorrect data point $x \in X$ is closest to $q$ under the fusion distance function of NHQ Wang et al. (2022; 2024).*

The fundamental issue with the NHQ fusion distance for our subset query constraint is that the *symmetric* label difference can lead us astray. Using the symmetric difference might be sufficient for a tabular label setup, where every base point has prescribed label values for a set of fixed attributes, and the query predicate is similarly set up, e.g., the base labels could have 2 attributes like `Venue` and `Year`, and the query could insist on `Venue=NeurIPS` and `Year=2024`. However, this formulation is insufficient for the MultiFilterANN problem where we merely want the label set $S_q$ of the query to be a subset of the label set of the base point $S_u$. As an example, we do not see a way to apply such an approach if a base vector has associated metadata like `Categories = {Sports, Baseball, Betting, NYTimes}` and the query simply wanted to filter by `Categories = {Betting, NYTimes}`. Rather, an *asymmetric* distance function on the label set is the right choice. This is elaborated in Section 6.1.

---

**Lesson 5:** A symmetric label set distance cannot capture the Multi-FilterANN problem.

---

## 6 OUR EMPIRICAL ALGORITHM

In this section, we describe our main empirical algorithm. It consists of three main parts:

- Building the graph index: we build our graph index using FilteredDiskANN from Gollapudi et al. (2023) with a modified greedy search routine, which we call PenaltyGreedySearch. For convenience, the full description of FilteredDiskANN can be found in Algorithm 2. The parameters of our build are given in our results; see Section 7.

- Searching the graph index: Motivated by our prior discussion of Section 5, we use a novel *penalized* search, described in Section 6.1. The full description is in Algorithm 1.

- Query planning: Based on 'Lesson 2', we route queries to either the graph search or simple search procedures depending on the selectivity of the query's labels. For example, if the query has a label which only one point in the base dataset satisfies, we can quickly check for this and just direct the query to the single relevant point. This is described in Appendix E.

## 6.1 Searching the Graph Index: The Penalty Graph Search

Most graph-based ANNS algorithms work in the following manner: the index construction involves building a directed graph $G$ with nodes corresponding to the base vectors; the search algorithm for a query vector employs a natural greedy or best-first traversal policy on $G$, by starting at some designated point $s \in P$, and iteratively hopping to the neighboring vertex whose vector is closest to the query vector until we reach a fixed point. Algorithm 1 with $\lambda = 0$ and $\tau = \infty$ gives a more formal definition of this process.

However, the drawback of such a search method is that we are completely agnostic to the query filters and base point labels. To overcome this, the FilteredDiskANN algorithm in Gollapudi et al. (2023) *only traverses the neighbors that satisfy the query predicate*. This change, along with a suitable change during the graph construction phase yielded reasonable results for simple predicates (like query having one filter label). However, as noted in Section 5, this strict approach is doomed to fail for very selective queries.

To overcome this deficiency, we note that the NHQ algorithm Wang et al. (2024) performs greedy search on a novel *fused distance*, which takes into account the distance $\|q - x\|$ between the query vector and the base vector $x$, and the symmetric difference between the label sets $S_q$ and $S_s$. However, Lemma 5.3 shows that this formulation also cannot capture the nuances of MultiFilterANN. Instead, we use an *asymmetric* distance on the label sets.

**Asymmetric Penalty Distance.** To provide intuition, the symmetric part of the NHQ fusion distance heavily penalizes data points that have many irrelevant labels, even if they contain all the labels of the query. Consider the following toy example: We have a universe of 5 labels, namely $\{A, B, C, D, E\}$, and there are are two base vectors $u$ and $v$, with label sets $S_u = \{A, B, C, D, E\}$ and $S_v = \{A, C\}$. Suppose we want to retrieve the closest vector to a query $q$ with label set $S_q = \{B\}$. Clearly, the correct result must return $u$ as it is the only feasible point. However, using a symmetric difference would result in returning $v$. Indeed, using the symmetric difference between $S_q$ and $S_v$ is essentially using the hamming (or squared euclidean, for that matter) distance between the one-hot encodings of the label sets, where the dimension of these encodings is of size $m$. Our simple-in-hindsight approach to solve this issue is to only measure the label difference *for the labels in $q$*.

This choice has three desirable properties. First, it does not penalize data points in $X$ that have many labels, skirting the lower bound presented in Lemma 5.3. Secondly, the asymmetric label distance can be embedded into $\ell_2$: represent the label set of the data points as the $\{0, 1\}$ valued $m$-dimensional vector where we have a coordinate of 1 iff the label is present. On the other hand, represent the label set of the query as a $\{1, 1/2\}$ valued vector, where again a coordinate of 1 is present iff the query has the label. The $\ell_2^2$ distance under this embedding *always* gives the same contribution for labels that are not present in the query. Thus, we essentially ignore labels not in the query when comparing the label sets of the query and a datapoint in $X$. Note that the asymmetry of the embedding is necessary to capture the asymmetric label set distance. As presented later, the fact that the asymmetric label distance can be embedded into $\ell_2$ implies that our final empirical algorithm has provable worst-case guarantees, under the common assumption of bounded doubling dimension (see Lemma D.1). Lastly, the fact that we can reduce the subset query part of MultiFilterANN to an $\ell_2$ distance calculation means it can be seamlessly combined with the vector search component.

Equipped with this, we can essentially reduce the MultiFilterANN problem to (vanilla) ANN by setting $\mathsf{dist}'(q, x) = \|q - x\|^2 + \lambda \cdot \|\mathsf{encF}(S_q) - \mathsf{encF}(S_x)\|^2$ where $\mathsf{encF}$ denotes the $m$-dimensional asymmetric encoding described above and $\lambda > 0$ is a large parameter.

**Theorem 6.1.** *Given a labelled set $X$ of bounded vectors, sufficiently large $\lambda$ and query vector $q$ with labels $S_q$, the closest-$k$ database vectors according to distance $\mathsf{dist}'$ is precisely the closest-$k$ feasible (i.e., those satisfying the label constraint of $S_q$) database vectors according to the original distance.*

While this establishes a rigorous connection between the filtered and unfiltered version of nearest-neighbor search, there are two issues we encounter when using it directly on real-world datasets: (a) performing distance computations over a $d + m$-dimensional vector space is significantly more expensive when the universe $U$ of labels is very large, as is often the case, and (b) each of the irrelevant labels in $U \setminus S_q$ contribute a value of $1/4$ (in squared euclidean metric) to the overall distance, which additively inflates all distances we are dealing with, and this consequently alters the behaviour of the algorithms used in practice. Due to these considerations, we settle on the following

asymmetric penalty distance which only cares about the labels in the query filter that are not present in the label set of the base point: $\widehat{\text{dist}}(q, x) = \|q - x\|^2 + \lambda|S_q \setminus S_x|$. In summary, once we have built the graph, we run the greedy search algorithm using the modified distance $\widehat{\text{dist}}$ defined above.

**Three Refinements** (a) One optimization to improve query latency without affecting recall much is to first evaluate the penalty distance $|S_q \setminus S_x|$, and only consider the candidate during search if this value is at most some threshold $\tau$. This is assuming that $S_q$ has small cardinality, and the set difference is significantly cheaper computationally compared to the distance calculation. (b) Second, our index data structure will also maintain a small sample set of the database points, as well as *label-wise inverted indices* on the sample set. That is, for each label $l \in U$, we let $\mathsf{sampList}(l)$ denote the set of database points in the sample that contain label $l$ in their filter sets. We start the greedy search from a suitable point which appears in the intersection of $\mathsf{sampList}(l)$ for $l \in S_q$. If the sample intersection has size 0 (which would happen when the filter predicates are highly selective), we simply start at the global starting point of the search. Our overall search algorithm is given in Algorithm 1. (c) Third, recall that in Gollapudi et al. (2023), FilteredDiskANN only considers candidate neighbors for a given point that has at least one matching filter. We find replacing this with the penalty method and a suitable value of $\tau$ provides better results.

## 7 EMPIRICAL RESULTS

We compare our algorithm from Section 6 to two open-source algorithms for ANNS with multiple filters: (a) Parlay$IVF^2$, the winning entry from the BigANN Filter competition Simhadri et al. (2022), and (b) ACORN, which claims state-of-art performance for this problem across a range of scenarios Patel et al. (2024). We remark that Zilliz and Pinecone have both released a proprietary algorithm accessible through the BigANN API. Both fail to run on our system, and are omitted from our comparisons. Finally, Irrespective of parameter choice, CAPS, another open-source algorithm gives near-zero recall on our primary datasets, and is therefore omitted from our comparisons.

In addition, we perform a number of ablation studies on our graph algorithm, demonstrating where it works well and where it struggles. All experiments are performed on a server with Intel(R) Xeon(R) Gold 5218 CPU at 2.30GHz, and 512 GB of RAM. We run all experiments with 1 thread.

### 7.1 DATASETS

We compare these algorithms on three diverse real-world datasets. Additionally, to the best of our knowledge, ours is the first work which uses entirely real-world datasets to evaluate algorithms for the MultiFilter-ANN problem, with all prior work partly or fully using datasets where the labels on which the queries are filtered are generated synthetically.

**Wiki-Cohere.** We introduce a dataset for AND query search based on the Cohere Wikipedia Embeddings dataset Cohere (2023). The dataset itself comprises of 35 million passages from Wikipedia pages. Each paragraph has an associated embedding as well as several pieces of metadata: the title of the page, the number of views, and the number of languages into which the page has been translated. Additionally, there is a simple English set with only 486,000 points. We use the embeddings as the base points. For the labels, we exclude "common words" (the NLTK stopwords Bird et al. (2009)), then take the 4000 most frequent words to be the label universe. Each embedding then uses the present words in the corresponding passage as labels. For the queries, we built a query set from the simple English set, with the filters being two random choices from the 15 most common words found in the corresponding passage. We avoid picking more than two choices to prevent queries from having a predicate with low selectivity; that is, have very few points in the index that satisfy the query's filters. We also use a 1M slice of this dataset in various experiments. There are 4000 labels in total, and the average number of labels per base point is 29.

**YFCC1M.** We also use a 1M slice of the YFCC dataset used in the BigANN Filter competition. The base vectors are 192-dimensional CLIP embeddings of images, and the queries are embeddings of texts. The metadata of the images, like camera model, resolution, etc. become the metadata based on which the query predicates are chosen (either a single label predicate or an AND of two labels). There are $\approx 182000$ labels, with the average number of labels per base point being 8.

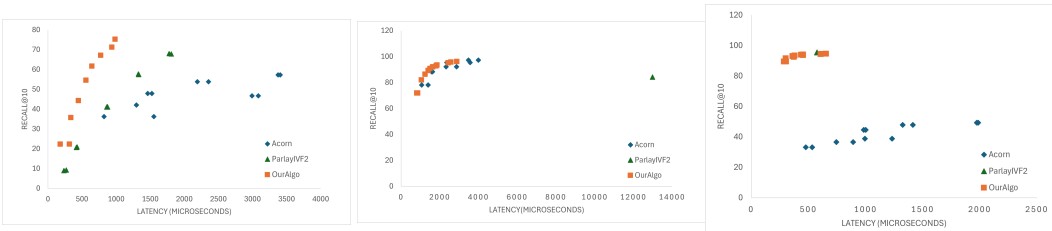

Figure 1: Recall vs Latency of different methods on Wiki-1M, Amazon-1M and YFCC-1M datasets.

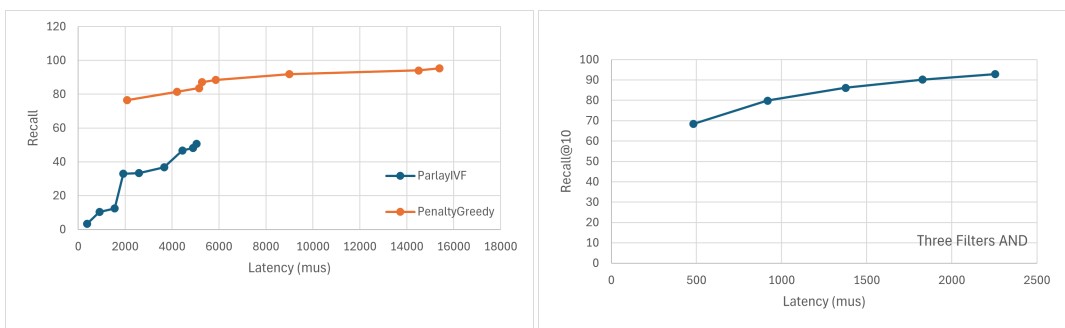

Figure 2: Comparison on Wikipedia-35M          Figure 3: Our algorithm on three filter ANDs

**Amazon.** We use a 1M slice of the Amazon dataset recently uploaded to the BigANN benchmarks. From Dataset (2024), the base vectors are $384$-dimensional Cohere embeddings of product descriptions, and the queries are embeddings of sample user queries generated by GPT. The predicates are based on user rating and product category (either a single predicate like Rating=5, or an AND query of the form CAT=Auto AND Rating=5). There are around $65000$ labels with an average of 6 labels per vector.

**Synthetic Datasets.** In addition to the above datasets, we also construct a series of datasets on top of the 1M slice of the Wikipedia-35 embeddings. The full procedures for generating these labels are detailed in the appendix. These datasets are designed for ablation studies on PenaltyDiskANN, and are not used for comparison with baselines due to having larger label sizes.

**Baseline Evaluations.** See Figure 1. We built the ACORN and Parlay$IVF^2$ indices with different parameter choices ($ef_C, \gamma, M$ for the former and cutoff and clusterSize for the latter) as suggested in the respective papers. We also build our graph index using $R = 64$, $L = 100$, and the FilteredVamana construction algorithm. For search, we sweep the $ef_S$ parameter for ACORN, tinyCutoff and targetPoints for ParlayIVF, and set bruteForceThreshold=10000 and clusteringThreshold=25000 for our algorithm. Interestingly, Parlay$IVF^2$ does not do well on the Amazon dataset, while ACORN exhibits some drawbacks on the YFCC dataset. We leave the task of understanding the reasons behind such contradictory performance of algorithms across datasets as future work.

We also compare our algorithm with Parlay$IVF^2$ on the full 35M wikipedia dataset and report the numbers in Figure 2. For the graph component of our algorithm, we set the build parameters of R= $64$ and L= $100$. For the clustering component, we use 1024 cluster centers. During search, we use a bruteforce search cutoff of 10000 and a clustering search cutoff of 50000. We also vary Ls from 10 to 80 in intervals of 10. For Parlay$IVF^2$, we set the build parameters cluster_size= 10000 and cutoff= 10000, the search parameters tiny_cutoff= 128000 and bitvector_cutoff= 128000, and vary target_points from 8000 to 20000 in increments of 4000. Finally, we show the generality of our algorithm by running it on queries with 3-term conjunctions (a AND b and C) as predicate on the wiki-1M dataset. Note that for comparable latency numbers, our algorithm offers a considerable recall improvement of around 40%.

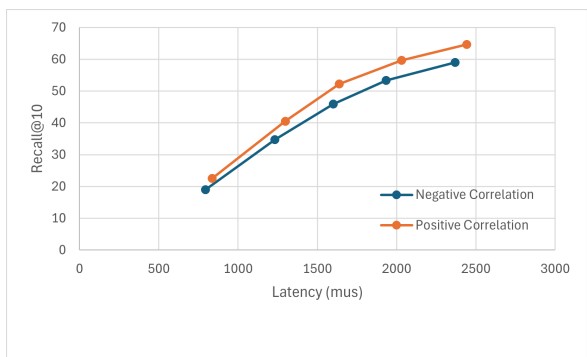

Figure 4: Query Correlation

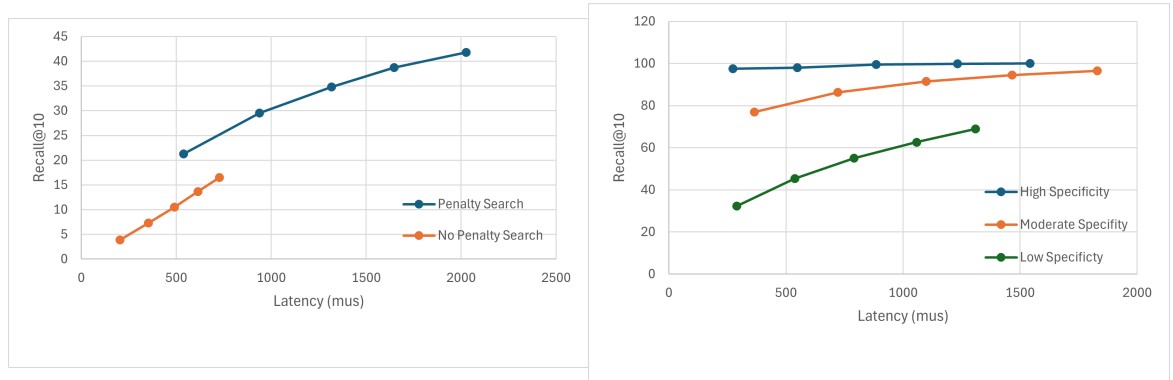

Figure 5: Penalty Removal

Figure 6: Query Filter Selectivity

## 7.2 ABLATIONS

We next investigate the effect of query correlation, filter selectivity, and the effect of the penalty term during graph search as ablation studies.

Query Correlation: Inspired by Patel et al. (2024), we develop synthetic labels that are correlated with the underlying embeddings to varying degrees. Intuitively, if a query vector has positively correlated labels, then we would expect its geometric nearest neighbors to generally the same labels. If a query has negatively correlated labels, then we would expect points far from the geometric nearest neighbors to have the labels. In the uncorrelated setting, we cannot make any assumptions about which points would have the labels with respect to the query's nearest neighbors. Figure 4 shows that for negatively, positively, and non-correlated query labels, there is a small change in performance in favor of positively correlated labels, suggesting that graph search performance is robust towards different query correlations.

Filter Selectivity: We evaluate for queries that come with filters of varying degrees of selectivity. Specifically, we study the case where queries come with high specificity labels (>90%), medium specificity labels (45%), and low specificity labels (10%). Figure 6 shows a considerable change in recall for lower selectivities. Indeed, if a query comes with rare filters that are evenly dispersed throughout the dataset, it becomes more difficult to find points with those filters using a distance-based greedy search, as is the case with our graph algorithm.

Removing the Search-time Penalty. As Figure 5 demonstrates, incorporating the penalty provides a minimum 2.5x increase to recall performance at the same QPS. The intuition here is that the penalty allows for an implicit prioritization of points with matching filters in the list $\mathcal{L}$ PenaltyGreedySearch maintains in its procedure. Without the prioritization, points that are close geometrically to the query without meeting the filter constraints can appear closer to the algorithm, leading to poor recall.

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

## A  SYNTHETIC LABEL GENERATION PROCEDURES

In all studies, a fixed random 1M slice of the Cohere Wikipedia dataset Cohere (2023) is used.

### A.1  FILTER SELECTIVITY STUDY LABELS

In this section, we provide the procedure to reproduce the labels generated for the experiment generated in Figure 6.

We partition a universe of 90 labels into equal sets of 30. Each partition has a different specificity, with the first having 90%, the second 45%, and the third having 10%. For each base point, we assign filters using the assigned selectivity as probabilities in a binomial trial for each partition. As such,

each point should have most of the first partition, half the second partition, and around 10% of the remaining partition as labels.

For the queries, we sample two filters from the partition corresponding to the selectivity we aim to evaluate.

### A.2 TRIPLE QUERY FILTER STUDY LABELS

We provide the procedure to reproduce the labels generated for the experiment shown in Figure 3.

We assign labels to each base point by performing a binomial trial on a universe of 60 filters with probability 0.5. For each query, we assign filters by sampling one filter from each sequential third (e.g. first 20 filters, second 20 filters, third 20 filters) of the universe.

### A.3 QUERY CORRELATION STUDY LABELS

In this section we describe the procedure for the construction of the synthetic label set used for the query correlation study (Figure 4).

First, we construct $k$ random separating hyperplanes. Each co-ordinate of each hyperplane is drawn from an independent gaussian mapping each datapoint to a "bucket" vector $\{0, 1\}^k$ (0 in the $i$th entry if it is below the $i$th hyperplane, 1 if it is above). This is similar to the procedure studied in Charikar (2002). For each label we randomly select 5 (out of $2^k$) bucket vectors as "foundations". Every datapoint in the foundation bucket has the label with probability 0.5. Every datapoint in an adjacent bucket has the label with probability 0.25 and other datapoints have the label with probability 0.01.

To make comparison more meaningful between the synthetic and natural datasets, we use the same 5000 query embeddings. Each of these queries has a corresponding bucket. We assign labels to the query based on the frequency of each label in the query bucket.

1. Positively correlated: Two distinct labels are chosen, each with probability proportional to the fraction of points in the bucket corresponding to the query satisfying the label.
2. Non-correlated: Two distinct labels are chosen uniformly at random.
3. Negatively correlated: Two distinct labels are chosen with probability proportional to the fraction of points *not* satisfying the label in the bucket corresponding to the query.

## B OMITTED DETAILS OF SECTION 4

### B.1 DETAILED OVERVIEW OF THE CIP DATA STRUCTURE

In this setting, we are given a universe of $m$ labels and a collection of $n$ label sets (data points). The goal is to construct a datastructure that takes as input a query set $q$ and outputs data points whose label sets are subsets of $q$.

The CIP-data structure comprises of 3 levels. The first level and the second level are both hash tables; the query is only routed to one entry in the first level and one entry in the corresponding second level table. The third level is a collection of disjoint sets $g$ of datapoints. It will turn out that (with high probability), for all $g$, either all datapoints in $g$ are valid responses or none are. However, we will need to evaluate the feasibility of a datapoint from each group. Fortunately, it will turn out that, for any $\delta \in (0, 1)$, we can construct such a structure with the number of such groups being at most $n^{1-\delta}$.

To construct the first level, we sample a random subset $S$ (with size $o(m)$) of the labels. The first level table will be indexed by the power set of $S$. The query $q$ will be mapped to $S' = q \cap S$. It turns out that, for each $S'$, with probability 0.5, we can construct a "representative" set $R$ such that for almost all datapoints $x$ satisfying $x \cap S \subset S'$, $|x - R|$ is small and $|R - q|$ is small. The second level table will be indexed by small subsets of $R$ (corresponding to $R - q$) and the third level table will have the datapoints grouped by $x - R$. More concretely:

1. We uniformly randomly sample a subset of the labels $S$ with size $k < m$ ($k$ depends on $\delta$). The table $T_1$ is indexed by subsets of $S$. A query $q$ will be mapped to the entry $S'$ such that $S' = q \cap S$. Corresponding to each entry $c_1$ of the table, we have the following:

- A list $L(c_1)$ of all datapoints $x$ such that $x \cap S \subset c_1$.
- A representative set $R(c_1) \subset \mathcal{L}$. This representative set is, loosely speaking, a "median" of the points in the $L(c_1)$. It will turn out that for most datapoints $x \in L(c_1)$, the set $x - R(c_1)$ is small. Note that $R(c_1)$ need not be a subset of $c_1$.
- A second level hash table $T_2(c_1)$ indexed by "small" subsets of $R(c_1)$.

2. $T_2(c_1)$ Second level table indexed by small subsets of $R(c_1)$. A query $q$ will be mapped to $T \subset R(c_1)$ if $T = R(c_1) - q$. Because the choice of $S$ in the first level table was random, with probability at least $0.5$ (for the right choice of $k$), $|R(c_1) - q|$ is small. If $R(c_1) - q$ is large, the data structure "fails". Corresponding to each cell $c_2$ of this table we have the following.

- A list $L(c_2)$ of datapoints $x$ such that $x \cap c_2 = \emptyset$.
- A partitioning $T_3(c_2)$ of the datapoints $x \in L(c_2)$ based on $x - R(c_1)$. Because for most datapoints, $x - R(c_1)$ is small, the number of components of the partition is small. From now on, we will call each component a "group".

3. $T_3(c_2)$, the partitioning of datapoints in $L(c_2)$. Once the query $q$ reaches $c_2$, we will check one datapoint $x(g)$ from each group $g$ in $T_3(c_2)$. If $x \subset q$, then every member of $g$ is a valid response to the query.

It's main guarantees are:

**Theorem B.1** (Charikar et al. (2002)). *For every $\delta > 0$, there exists a data structure for the SubsetQuery problem that has $2^{O(m\sqrt{\delta} \log^2 m)}$ cells, each of which contains at most $O(n^{1-\delta})$ disjoint groups of elements and, with probability at least $0.99$, maps any query to a single cell, returning the $O(n^{1-\delta})$ groups of elements. Either all the elements in a group are valid for the query or none are.*

We note that the space of the data structure is only subexponential when $\delta = o(\frac{1}{\log^4 m})$. No algorithm for the SubsetQuery problem is known that achieves subexponential storage space when $\delta$ is a constant. Furthermore, $n^{1-\delta} = o(n)$ for a wide range of choices for $m$. Indeed, we have $n^{1-\delta} = n/\text{poly} \log(n) = o(n)$ up to $m = \exp\left((\log(n)/\log\log n)^{O(1)}\right)$, i.e., barely sub-polynomial.

Finally we note that the $0.99$ success probability can be boosted to an arbitrary value close to $1$ by standard independent repeating.

### B.2 PROOF OF THEOREM 4.1

First we review Locality Sensitive Hashing (LSH). It has the following guarantees, parameterized by the value $\rho$:

**Theorem B.2** (LSH, Indyk & Motwani (1998a)). *Let $d(\cdot)$ be a distance function. Let $p_1$ and $p_2$ be such that for every $r > 0$, there exists a distribution over hash functions $f$ such that for every $x, y$, $Pr(f(x) = f(y)) \geq p_1$ if $d(x, y) \leq r$ and $Pr(f(x) = f(y)) < p_2 < p_1$ if $d(x, y) \geq (1+\varepsilon)r$. Then letting $\rho := \frac{\log(p_1)}{\log(p_2)}$, there exists a data structure with storage space $\tilde{O}(n^{1+\rho} + nd)^2$ which can find a $(1+\varepsilon)-$approximate nearest neighbor while performing at most $\tilde{O}(n^\rho)$ distance comparisons.*

In the case where the distance function is the Euclidean metric ($\ell_2$), prior works achieve $\rho = 1 - \varepsilon$ Andoni et al. (2018).

**Theorem 4.1.** (Main theoretical guarantee for Multi-FilterANN) *There exists a data structure which, with probability $0.99$, returns a $(1+\varepsilon)-$approximate filtered nearest neighbor on any query. This data structure uses $\tilde{O}(n^{1-\delta}(m + dn^\delta + n^{3\delta})2^{O(m\sqrt{\delta}\log^2 m)})$ space and on any query, performs at most $O(n^{1-\delta})$ set intersections and $\tilde{O}(n^{1-\varepsilon\delta})$ distance comparisons.*

*Proof.* First we focus only on the labels. Each base point corresponds to a subset of the labels; multiple points may correspond to the same subset of labels.

---

$^2\tilde{O}(\cdot)$ hides logarithmic factors.

We construct the same data structure as in Theorem B.1. Next, for every cell of the level two table, we take groups with more than $n^\delta$ points and split them into smaller groups containing $n^\delta$ points each (one of the groups created by each split may have fewer than $n^\delta$ points).

The original structure had at most $n^{1-\delta}$ groups per cell of the second level table. Post splitting, there will be at most $n^{1-\delta}$ groups of size $n^\delta$ and at most $n^{1-\delta}$ smaller groups, $2n^{1-\delta}$ groups of size at most $n^\delta$.

Next, in addition to having a list of points in each group, we give each group an Locality Sensitive Hashing based structure (Theorem B.2) on only the datapoints in that group. The search for each group uses at most $\tilde{O}(n^{\delta\rho})$ distance comparisons.

Since there are $O(n^{1-\delta})$ groups, the total number of distance comparisons is at most $\tilde{O}(n^{\delta\rho}n^{1-\delta}) = \tilde{O}(n^{1-(1-\rho)\delta})$.

We know from Theorem B.1 that the total number of cells in all level 2 tables is at most $2^{O(m\sqrt{\delta}\log^2 m)}$. Each cell contains at most $O(n^{1-\delta})$ groups of size at most $n^\delta$. Each LSH structure requires storage space at most $O(dn^\delta + n^{\delta+\delta\rho})$ and each group also needs to store a list points and the groups' difference to the representative, which is $\tilde{O}(m + n^\delta)$. Therefore, the total storage space for the group is at most $\tilde{O}(n^{1-\delta}(m + dn^\delta + n^{\delta+\rho\delta})2^{O(m\sqrt{\delta}\log^2 m)})$ as desired. Recall that we can take $\rho = 1 - \varepsilon$. $\qquad\square$

**Corollary B.3.** *Given a data structure for the unfiltered approximate nearest neighbor problem with storage space $S(n, d, \varepsilon)$ and query time $Q(n, d, \varepsilon)$, we can, by replacing the groups in CIP with the given unfiltered data structure to obtain a storage space of $\tilde{O}(n^{1-\delta}(m + S(n^\delta, d))2^{O(m\sqrt{\delta}\log^2 m)})$ and query time $O(n^{1-\delta}(Q(n^\delta, d) + m))$.*

**Remark B.1.** *The 'slow preprocessing version' of) DiskANN Indyk & Xu (2023) has space $S(n, d, \varepsilon) = n \cdot (1/\varepsilon)^{O(\lambda)}$ and query time $Q(n, d, \varepsilon) = \tilde{O}((1/\varepsilon)^{O(\lambda)})$, where $\lambda$ is the doubling dimension of the (vector) part of the dataset $X$. We can apply this algorithm to Corollary B.3. From our theoretical algorithm, we retain two valuable lessons that we use in our empirical algorithm design.*

## C    OMITTED PROOFS OF SECTION 5

**Lemma 5.1.** *There exists a size-$n$ one-dimensional dataset with two total labels such that the ParlayIVF$^2$ algorithm has query time $\Omega(n)$.*

*Proof.* The instance we construct will be 1 dimensional and points will have 2 labels. We will have $n$ points and assume for simplicity that $n$ is odd. We have points in the odd positions $1, 3, 5 \ldots n - 2$ with label 1 and points in the even positions $2, 4, 6 \ldots n - 1$ with label 2 and a point in position $n$ with both labels 1 and 2. The query will be at location 0 and contain both labels 1 and 2. If Parlay has $k$ clusters containing label 1 and $k'$ clusters containing label 2 for some $k, k'$, then each cluster will correspond to all the points containing a label in an interval. The clusters containing the point in position $n$ will be the last clusters added to the queue in Parlay. Therefore, when performing the intersection, Parlay will have to intersect two sets of size $\Omega(n)$ and therefore, Parlay must have either linear running time or worst case recall of 0. $\qquad\square$

**Lemma 5.2.** *There exist a labeled dataset $X$ of size $n$ with $m = O(\log n)$ total labels such that any graph index on $X$ with the property that the subgraph of points satisfying the label constraints of a query is connected, must have $\Omega(n^2)$ edges.*

*Proof.* We set $m = C\log n$ for a sufficiently large constant $C$. For each label $\ell \in [m]$ and datapoint $x \in X$, assign label $\ell$ to $x$ with probability $\frac{1}{4}$ (independently of all other label-datapoint pairs). Notice that if two datapoints have label sets $A, B$ and there is no other datapoint whose label set contains $A \cap B$, then there must be an edge between the two points with label sets $A$ and $B$ (if the query is $A \cap B$ then $A$ and $B$ are the only valid responses so there must be an edge between them).

Let $X_C^{AB\ell}$ be 1 if label $\ell$ is in $A$ and $B$ but not $C$ and 0 otherwise. If the edge $A, B$ is not in the graph, then there exists a $C$ such that $X_C^{AB\ell} = 0$ for all labels $\ell$. For each label $k$, the probability

that $k$ is in $A \cap B$ but not $C$ is (by independence of the assignments) $\frac{1 \times 1 \times 3}{4 \times 4 \times 4} = \frac{3}{64}$. Therefore (by independence of the assignments) the probability that $X_C^{AB\ell} = 0$ for all labels $\ell$ is $(\frac{61}{64})^m$.

For any pair of points $A$, $B$, there are $n - 2$ possible datapoints that contain the entire intersection. The probability that none of them do is (by a union bound) at least $1 - (n-2)(\frac{61}{64})^m$. If $m \geq \frac{\log(100(n-2))}{\log \frac{64}{61}} = \frac{\log n}{\log \frac{64}{61}} + \frac{\log 100}{\log \frac{64}{61}}$, then this probability is at least $(1 - \frac{n-2}{100(n-2)})$, which is 0.99. This means every pair of points has an edge between them with probability at least 0.99 and so the expected degree of any node is at least $0.99(n-1) \geq 0.5n$ as desired. $\qquad\square$

**Lemma 5.3.** *There exists a one-dimensional dataset $X$ and a query $q$ such that an incorrect data point $x \in X$ is closest to $q$ under the fusion distance function of NHQ Wang et al. (2022; 2024).*

*Proof.* Suppose we are in $\mathbb{R}^1$ and a query has label set $\{a, b\}$ at position 0. There are two points in $X$, one with label set $\{a, b, c\}$ at position 1 and one with label set $\{a\}$ at position 0. Applying the NHQ method will favor the latter point (same distance with respect to the label set for NHQ, but closer geometrically). However, the correct solution for the subset query would be the one with label set $\{a, b, c\}$. $\qquad\square$

## D    MISSING DETAILS OF SECTION 6

**Lemma D.1.** *Let $m_X$ and $m_L$ be the doubling dimensions of $X$ and of the set of label indicator vectors with the $\ell_2$ metric, respectively. Define the metric on their product as $\widehat{\text{dist}}(x, x') = \|x - x'\|_2^2 + \lambda \cdot \|S_x - S_{x'}\|_2^2$. Then the resulting metric space has doubling dimension at most $2(m_X + m_L)$. Moreover, if $\lambda$ is large enough, then the doubling dimension is at most $\max(m_X, 2m_L)$.*

*Proof.* Let $L = \{S_x : x \in X\}$; we treat $X$ and $L$ as two $\ell_2$ metric spaces. Fix any $(x, \ell) \in X \times L$ and $r > 0$. Applying the assumption twice, we can obtain a collection of balls $B_X(x_i, \frac{r}{2})$ for $i = 1, \ldots, 2^{2m_X}$ that cover $B_X(x, 2r)$, and a collection of balls $B_L(S_j, \frac{r}{2\sqrt{\lambda}})$ for $j = 1, \ldots, 2^{2m_L}$ that cover $B_L(\ell, \frac{2r}{\sqrt{\lambda}})$. We claim that in the metric space $(X \times L, f)$ where $f^2((x, \ell), (x', \ell')) = \|x - x'\|_2^2 + \lambda \cdot \|\ell - \ell'\|_2^2$, the collection of $2^{2m_X + 2m_L}$ balls

$$\{B_f((x_i, S_j), r) : i = 1, \ldots, 2^{2m_X}, j = 1, \ldots, 2^{2m_L}\}$$

covers $B_f((x, \ell), 2r)$. (Note that we are using $f^2$ instead of $\widehat{\text{dist}}$ for ease of notation since we make use of $f$). To show this, take any $(x', \ell') \in B_f((x, \ell), 2r)$, i.e., $\|x - x'\|_2^2 + \lambda \cdot \|\ell - \ell'\|_2^2 \leq 4r^2$. Then $\|x - x'\|_2 \leq 2r$ and $\|\ell - \ell'\|_2 \leq \frac{2r}{\sqrt{\lambda}}$, so there are some $i$ and $j$ such that $x' \in B_X(x_i, \frac{r}{2})$ and $\ell' \in B_L(S_j, \frac{r}{2\sqrt{\lambda}})$. Thus $f^2((x', \ell'), (x_i, S_j)) \leq \frac{r^2}{4} + \lambda \cdot \frac{r^2}{4\lambda} = \frac{r^2}{2} \leq r^2$, i.e., $(x', \ell') \in B_f((x_i, S_j), r)$.

For the second point, intuitively, if $\lambda$ is large enough, then the metric space $(X \times L, f)$ consists of clusters $X \times \{\ell\}$ that are well-separated. Concretely, let $\lambda \geq \frac{16}{3}\text{diam}^2(X)$. Now fix any $(x, \ell) \in X \times L$ and $r > 0$ and consider the ball $B_f((x, \ell), 2r)$. We have two cases.

If $2r < \sqrt{\lambda}$, then (as $\|\ell - \ell'\|_2^2 \geq 1$ for $\ell' \neq \ell$) the entire ball is contained in $X \times \{\ell\}$, and restricted to that set, $f = \|\cdot\|_2$. Therefore we can take a collection of balls $B_X(x_i, r)$ for $i = 1, \ldots, 2^{m_X}$ that cover $B_X(x, 2r)$, and lift them to $B_f((x_i, \ell), r)$.

If $2r \geq \sqrt{\lambda}$, cover $B_L(\ell, \frac{2r}{\sqrt{\lambda}})$ with a collection of balls $B_L(S_j, \frac{r}{2\sqrt{\lambda}})$ for $j = 1, ..., 2^{2m_L}$. We claim that the collection $B_f((x, S_j), r)$ for $j = 1, \ldots, 2^{2m_L}$ covers $B_f((x, \ell), 2r)$. To show this, take any $(x', \ell') \in B_f((x, \ell), 2r)$, i.e., $\|x - x'\|_2^2 + \lambda \cdot \|\ell - \ell'\|_2^2 \leq 4r^2$. Then $\|\ell - \ell'\|_2 \leq \frac{2r}{\sqrt{\lambda}}$, so there is $j$ such that $\ell' \in B_L(S_j, \frac{r}{2\sqrt{\lambda}})$, i.e., $\lambda \cdot \|\ell' - S_j\|_2^2 \leq \frac{r^2}{4}$. Moreover, $\|x - x'\|_2^2 \leq \text{diam}^2(X) \leq \frac{3}{16}\lambda \leq \frac{3}{4}r^2$. In total, $(x', \ell') \in B_f((x, S_j), r)$. $\qquad\square$

The above lemma implies that we can search using our distance $\widehat{\text{dist}}$ on graph indices whose runtimes depend on the doubling dimension, for example the DiskANN analysis of Indyk & Xu (2023).

**Theorem 6.1.** *Given a labelled set $X$ of bounded vectors, sufficiently large $\lambda$ and query vector $q$ with labels $S_q$, the closest-$k$ database vectors according to distance $\mathsf{dist}'$ is precisely the closest-$k$ feasible (i.e., those satisfying the label constraint of $S_q$) database vectors according to the original distance.*

*Proof.* If $\lambda$ is larger than say twice the diameter of the vectors in $X$ (for example larger than 2 for normalized unit vectors), then the second term of $\mathsf{dist}'$ dominates. Thus, any base vector $x$ that does not satisfy the label constraint (i.e., $S_q$ is not a subset of $S_x$), will have $\mathsf{dist}'$ value larger than all the vectors in our dataset that satisfy the label constraint. □

## E  QUERY PLANNING

While the main contribution in this work is in making graph-based indices more robust to handling query predicates, as discussed earlier, almost any graph-based search algorithm would struggle when the query predicate is highly selective. Indeed, in extreme situations where a very small number of points satisfy the predicate, it might just make sense to first identify these points (say, by intersecting inverted indices of each of the query labels), and running brute force distances to compute the closest $k$ points. It turns out that for our algorithm in its current shape, there are also some predicates of low, but not too low, selectivity for which an intermediate clustering-based data-structure works best.

Given the labelled dataset $X$ of vectors, we choose a target numClus of the number of clusters, and run a $k$-means clustering with $k = \mathsf{numClus}$ to generate the clusters. For each cluster $C_i \subset X$, we maintain label-wise inverted indices $\mathsf{invList}(C_i, l)$ that contains the ids of database points that fall into cluster $C_i$ and have label $l$ in their label sets. When the query $q$ arrives with label set $S_q$, we first identify the closest $L$ clusters, say, $C_1, C_2, \ldots, C_L$ from the numClus clusters (using brute force search). Then, for each of these $L$ clusters $C$, we compute $\cup_{i=1}^{L} \cap_{l \in S_q} \mathsf{invList}(C_i, l)$ to gather the feasible points in these clusters, and then identify the closest $k$ points to the query, again via brute force search, from this set to generate our output.

Putting all these pieces together, our final empirical algorithm is then defined as follows: Given the query $q$ with labels $S_q$, estimate the fraction of database points which will satisfy $S_q$ using a sample dataset along with label-wise inverted indices over the sample.

1. If the estimate is tiny (e.g. 10000 points), we use a brute-force search.

2. If the estimate is moderate (e.g. 50000 points), we search using the clustering layer.

3. If the estimate is large, we run the penalty greedy search.

---

**Algorithm 1:** PenaltyGreedySearch($x_q, S_q, k, s, L, \lambda, \tau$)

---

**Data:** Query vector $q$, query filter(s) $S_q$, start node $s$, search list size $L$, and penalty parameters $\lambda$ and $\lambda$.
**Result:** Result set $\mathcal{L}$, and a set $\mathcal{V}$ containing all visited nodes.
**begin**

   1     For any $u$, define $f(u) := \|x_u - q\| + \lambda|S_u \setminus S_q|$

   2     Initialize sets $\mathcal{L} \leftarrow \{s\}$ and $\mathcal{V} \leftarrow \emptyset$.

          **while** $\mathcal{L} \setminus \mathcal{V} \neq \emptyset$ **do**

   3         Let $p^* \leftarrow \arg\min_{p \in \mathcal{L} \setminus \mathcal{V}} f(p)$

   4         $\mathcal{V} \leftarrow \mathcal{V} \cup \{p^*\}$

   5         Let $N'_{\mathrm{out}}(p^*) \leftarrow \{p' \in N_{\mathrm{out}}(p^*) : |S_{p'} \setminus S_q| < \tau\}$

   6         $\mathcal{L} \leftarrow \mathcal{L} \cup N'_{\mathrm{out}}(p^*)$

             **if** $|\mathcal{L}| > L$ **then**

   7             Update $\mathcal{L}$ with the closest $L$ nodes to $x_q$ with respect to $f$.

         **return** [Closest $k$ NNs from $\mathcal{L}$ satisfying $S_q$; $\mathcal{V}$]

---

---

**Algorithm 2:** FilteredDiskANN Indexing Algorithm

---

**Data:** Database $P$ with $n$ points where $i$-th point has coords $x_i$, parameters $\alpha, L, R$.
**Result:** Directed graph $G$ over $P$ with out-degree $\leq R$.
**begin**

1    Initialize $G$ to an empty graph
2    Let $s$ denote the medoid of $P$
3    Let $\mathsf{st}(f)$ denote the start node for filter label $f$ for every $f \in F$
4    Let $\sigma$ be a random permutation of $[n]$
5    Let $F_x$ be the label-set for every $x \in P$
   **foreach** $i \in [n]$ **do**
6      Let $S_{F_{x_{\sigma(i)}}} = \{\mathsf{st}(f) : f \in F_{x_{\sigma(i)}}\}$
7      Let $[\emptyset; \mathcal{V}_{F_{x_{\sigma(i)}}}] \leftarrow \mathtt{FilteredGreedySearch}(S_{F_{x_{\sigma(i)}}}, x_{\sigma(i)}, 0, L, F_{x_{\sigma(i)}})$
8      $\mathcal{V} \leftarrow \mathcal{V} \cup \mathcal{V}_{F_{x_{\sigma(i)}}}$
9      Run $\mathtt{FilteredRobustPrune}(\sigma(i), \mathcal{V}_{F_{x_{\sigma(i)}}}, \alpha, R)$ to update out-neighbors of $\sigma(i)$.
     **foreach** $j \in N_{\text{out}}(\sigma(i))$ **do**
10        Update $N_{\text{out}}(j) \leftarrow N_{\text{out}}(j) \cup \{\sigma(i)\}$
       **if** $|N_{\text{out}}(j)| > R$ **then**
11          Run $\mathtt{FilteredRobustPrune}(j, N_{\text{out}}(j), \alpha, R)$ to update out-neighbors of $j$.

---