# OpenReview forum: "Graph-based algorithms for nearest neighbor search with multiple filters"
_ICLR.cc/2025/Conference — ICLR 2025 Conference Withdrawn Submission_

### Official Review · Reviewer_j3JW · 2024-10-25

**Soundness:** 2
**Presentation:** 2
**Contribution:** 2
**Rating:** 3
**Confidence:** 4

**Summary:**

The paper studies the problem called MultiFilterANN (MFANN for short) that extends the approximate nearest neighbor search (ANNS) with additional constraints.
Such new constraints require a new label set Sx for each data point x, and ask to answer ANNS given a query q and label set Sq, such that Sq \subset Sx for the returned top-k nearest neighbors x.
This problem is of important in many applications in recommender system and vector search engines.

The paper first proposes to study the Subset Query problem (SQ for short) to deal with the constraint Sq \subset Sx.
In theory, the paper shows that combining the solution of SQ with popular (LSH-based or graph-based) data structures can solve MFANN, and points out that the complexity of MFANN is dominated by SQ (Theorem 4.1 and 4.2).
In practice, the paper shows a variant of graph-based data structure, called FilterDiskANN, with a new fused distance measure incorporating the contrainsts via a new embedding.
Empirical results show that the proposed method is competitive with ACORN and clustering-based ParlayIVF2 in many real-world data sets.

**Strengths:**

S1. The paper studies an emerging problem MultiFilterANN (MFANN), a variant of ANNS with attribute contraints, which is important in many LLM-based applications and vector search engines.

S2. A theoretical analysis of the subset query (SQ) problem to MFANN is proposed.

S3. A practical graph-based solution to answer MFANN is proposed.

**Weaknesses:**

W1. On theoretical side, the SQ problem only determines that there exists Sx such that Sq \subset Sx. It does not report **all** Sx such that Sq \ subset Sx. Therefore, I think the claim of Theorem 4.1 and 4.2 are not correct.

A simple case for LSH is follows.
Assume that there exists an cr instance x that is not in the reported group of the SQ's data structure, then LSH will report empty at the certain radius r. The algorithm then continues with larger c. In this case, you cannot guarantee (1+ \eps)-ANNS.
Another sanity check is that all instances x \in X satisfies the constraint. Hence, reporting a group of n^\delta points by CIP (Charikar et al. 02) clearly affects the (1+\eps) guarantees.
Since SQ's data structure cannot guarantee to report all elements x such that Sq \subset Sx, there will be an uncontrolled probability that DIC & LSH fail to guarantee (1 + \eps)-ANNS. Unfortunately, I do not think the used technique CIP & LSH (or graph-based index) can fix this issue without increasing the time and space complexity.

W2. I found the theoretical analysis and the proposed algorithms are not aligned well though there are several raised issues (i.e. lessons) to connect them together. In the end, the proposed algorithm is a combination between FilteredDiskANN (Gollapudi et al. 2023) with new fused distance measure (inherited in NHQ - Wang et al. 2023).

W3. Since the proposed graph-based data structure is very similar to FilteredDiskANN and NHQ, these methods should be used as baselines. Unfortunate, there are no comparison with these works in the experiment sections. The key parameter \lambda used in the proposed penalty distance has not been investigated (the value setting, sensitivity).

W4. There are two new fused distances including dist' and \hat{dist}, which use Sq in the querying phase. Though it is not clear to me which measure are used for querying, it seems that the proof of Theorem 6.1 with dist' in the appendix is not correct.
Given a large \lambda, the contribution of second term dominates dist'. How could it guarantee to answer MFANN according the distance in the first term? It looks to me the proof is an heuristic argument since given the graph index, filtering out point y such that Sq \notsubset Sy cannot bring you to the closet point x such that Sq \subset Sx.

W5. There are significant room to improve the paper.

For example, Related work need to be rewritten.
First half of the content is just listing the citations without any discussions.
The second half includes only claims. E.g.
CAPS seems to suffer significantly as m increase --> By how it suffers?
SERF works with range queries --> MFANN is not designed for range queries as well
These ideas do not generalize to multi filters --> Reasons?
ACORN do not scale well to even simple AND predicates --> By how?

Given previous works on graph-based solutions (ACORN, FilteredDiskANN, NHQ) for MFANN, I do not think all raised lessons are important. Perhaps sticking on the key lesson directly influenced the design of your final algorithm might be better.

**Questions:**

Could you address the raised weaknesses W1 -- W4 above?

There are several claims on the paper that seem to be vague or need the contexts. So I raise following questions for clarification

Q1) What is the definition of space-time tradeoff of MultiFilterANN?
Q2) What are lower bounds of space or time complexity of algorithms for MultiFilterANN?
Q3) In the query phase, the algorithm only uses \hat{dist}. The proposed dist' is only for theoretical interest. Is this correct?
Q4) Could you report a sensitivity analysis for the λ parameter?

---

> ### Author Response · Authors · 2024-11-26
>
> We thank the reviewer for the detailed response and concerns. Below, we address the reviewer's listed weaknesses in our submission (split over multiple comments):
>
> > W1: On theoretical side, the SQ problem only determines that there exists $S_x$ such that $S_q \subset S_x$. It does not report all $S_x$ such that $S_q \ subset S_x$. Therefore, I think the claim of Theorem 4.1 and 4.2 are not correct.
>
> > A simple case for LSH is follows. Assume that there exists an cr instance $x$ that is not in the reported group of the SQ's data structure, then LSH will report empty at the certain radius $r$. The algorithm then continues with larger $c$. In this case, you cannot guarantee $(1+ \epsilon)$-ANNS. Another sanity check is that all instances $x$ \in $X$ satisfies the constraint. Hence, reporting a group of $n^\delta$ points by CIP (Charikar et al. 02) clearly affects the $(1+\epsilon)$ guarantees. Since SQ's data structure cannot guarantee to report all elements $x$ such that $S_q \subset S_x$, there will be an uncontrolled probability that DIC & LSH fail to guarantee $(1 + \epsilon)$-ANNS. Unfortunately, I do not think the used technique CIP & LSH (or graph-based index) can fix this issue without increasing the time and space complexity.
>
> The reviewer may have missed a crucial definition. We do not require reporting *all* $x$ such that $S_q \subseteq S_x$. Rather, we need to report one $x$ that satisfies this guarantee, under the constraint that the $\ell_2$ distances are also minimized.
>
> However, we note that we can easily turn our data structure to reporting all such $x$ such that $S_q \subseteq S_x$. The CIP datastructure actually does, with probability 0.5, return a pointer to groups such that every single feasible point for the query is in some group. Of course, the groups may collectively contain every single point, but performing ANN in sublinear time on each group enables us to find a solution in $o(n)$ time. We will edit the paper to make this clearer. Moreover, we stress that all prior work on multifiltered-NN has been purely empirical in nature. We provide the first meaningful theoretical guarantees for this problem and show that we can achieve almost the same space-time tradeoffs as the best known for subset query.

---

> ### Author Response · Authors · 2024-11-26
>
> > W2: I found the theoretical analysis and the proposed algorithms are not aligned well though there are several raised issues (i.e. lessons) to connect them together. In the end, the proposed algorithm is a combination between FilteredDiskANN (Gollapudi et al. 2023) with new fused distance measure (inherited in NHQ - Wang et al. 2023).
>
> We thank the reviewer for the feedback. The lessons learned from our theoretical algorithm (lessons 1-4) directly translate into our empirical algorithm design process.
>
> Lesson 1 is saying graph based algorithms can also be used for our main theoretical result of Theorems 4.1 and 4.2. Thus, we do not lose any theoretical power by using graph based approaches.
>
> For Lesson 2, query planning refers to appropriately routing the query to a small part of the dataset so that we can guarantee a subset match exists while only comparing to a tiny fraction of data for efficiency. Both the theoretical algorithms and our practical variant make use of this meta-idea.
>
> Lastly, Lesson 3 highlights that popular prior approaches for our problem, which use clustering based methods, are too ‘local.’ In a simple one dimensional example, clustering based methods spend too much time focusing on the ‘distance’ part of the problem and ignore the subset match constraint. This forces them to iterate over many irrelevant points in the local vicinity of the query. In contrast, graph based methods allow us to skip many local points automatically since we consider both the distances and subset matches together.

---

> ### Author Response · Authors · 2024-11-26
>
> > W3: Since the proposed graph-based data structure is very similar to FilteredDiskANN and NHQ, these methods should be used as baselines. Unfortunate, there are no comparison with these works in the experiment sections. The key parameter \lambda used in the proposed penalty distance has not been investigated (the value setting, sensitivity).
>
> While our method does use ideas from these works, both FilterDiskANN and NHQ *cannot handle* the multi-filter problem. They are designed for very different usecases: FilterDiskANN can only handle one label. Of course, we can reduce the multi-filter problem to the one filter setting, this incurs an exponential blowup in the number of labels, which is prohibitive and impractical for all of our datasets.
>
> NHQ on the other hand can easily be shown to give incorrect answers; we kindly refer the reviewer to our lower bound in Line 286.

---

> ### Author Response · Authors · 2024-11-26
>
> > W4. There are two new fused distances including $dist'$ and $\hat{dist}$, which use $S_q$ in the querying phase. Though it is not clear to me which measure are used for querying, it seems that the proof of Theorem 6.1 with $dist'$ in the appendix is not correct. Given a large $\lambda$, the contribution of second term dominates $dist'$. How could it guarantee to answer MFANN according the distance in the first term? It looks to me the proof is an heuristic argument since given the graph index, filtering out point $y$ such that $S_q \notsubset S_y$ cannot bring you to the closet point $x$ such that $S_q \subset S_x$.
>
> We thank the reviewer for the comment. The $\lambda$ term has the same value ($C$) for all datapoints that are feasible for the query. Moreover, for any infeasible point, the lambda term is larger than ($C+$Diameter). Therefore, any nearest neighbor with the new distance is a nearest feasible neighbor in vanilla distance.

---

> ### Author Response · Authors · 2024-11-26
>
> > W5. There are significant room to improve the paper.
>
> > For example, Related work need to be rewritten. First half of the content is just listing the citations without any discussions. The second half includes only claims. E.g. CAPS seems to suffer significantly as m increase --> By how it suffers? SERF works with range queries --> MFANN is not designed for range queries as well These ideas do not generalize to multi filters --> Reasons? ACORN do not scale well to even simple AND predicates --> By how?
>
> We thank the reviewer for the feedback on our related works section. We will include an expanded discussion of related approaches and clean up our related works section in our final submission.

---

> ### Author Response · Authors · 2024-11-26
>
> We also thank the reviewer for the additional questions, and address them below.
>
> > Q1) What is the definition of space-time tradeoff of MultiFilterANN?
>
> Intuitively, the less storage space you have for your datastructure, the less powerful the structure is and the worse the query time will be. Therefore, the spacetime tradeoff can be defined as the curve of possible query times for a given amount of storage space.
>
> > Q2) What are lower bounds of space or time complexity of algorithms for MultiFilterANN?
>
> Since the problem is a generalization of subset query, the theoretical guarantees we give almost match the best known for subset query. Improving further on these has been a significant open problem for decades.
>
> > Q3) In the query phase, the algorithm only uses $\hat{dist}$. The proposed $dist'$ is only for theoretical interest. Is this correct?
>
> During search, we only use the hat distance.
>
> > Q4) Could you report a sensitivity analysis for the λ parameter?
>
>  Lambda just needs to be larger than the diameter of the dataset, as the proof of Theorem 6.1 shows.

---

> ### Comment · Reviewer_j3JW · 2024-11-26
> **Your argument does not fix the issues**
>
> Thank for detailed feedbacks.
>
> Regarding W1 and W4:
>
> From what I understand, Theorem 4.1 and 4.2 are both for worst-case scenarios. My raised issue is that if the found x from CIP does not satisfy the filtering condition but there is another y that satisfies the filtering condition, then your theoretical guarantees are not correct. If you cannot find all points (e.g. x and y) from CIP then you cannot guarantee any (1 + \eps)-ANNS.
>
> If you search for all points, then the running time of CIP component will be the bottleneck, and I do not think applying LSH on each CIP group will reduce the total running time to $o(n)$.
>
> Even \lambda is fixed for every point, the Euclidean squared distance $|q - x|^2$ varies for each points, so I do not think your argument is convincing.
>
> Given the raised weakness (potential errors) in theory and lack of comparison with key competitors (e.g. NHQ - in practice the proposed solution is similar to NHQ, only different from the heuristic fused distance), I maintain the rating.

---

### Official Review · Reviewer_dpjP · 2024-10-30

**Soundness:** 4
**Presentation:** 3
**Contribution:** 4
**Rating:** 8
**Confidence:** 3

**Summary:**

The paper studied the high-dimension nearest neighbor search problem with label constraints. The problem is defined as follows: we are given a dataset $X$ where each item is a vector and has a set of labels; we are required to build a data structure such that for every given query vector $q$ with a set of labels $S_q$, we should find a vector $v\in X$ that is approximately the closest and the labels $S_v$ is a subset of $S_q$. The problem is motivated by real-life search and recommendation applications, where we are usually required to return ‘similar queries’ with some properties specified by ‘filters’.

There are both theoretical and practical results in this paper. On the theoretical side, the paper designed an algorithm that uses $o(2^m)$ memory and returns an approximation solution in $o(n)$ time, where $n$ is the number of elements in the dataset and $m$ is the maximum cardinality of $S_q$. The algorithm is based on the subset-query algorithm of CIP [ICALP’02] and the celebrated LSH. Furthermore, by substituting the LSH with graph-based structures, the paper obtained dimensionality-independent memory bound that scales with $1/\varepsilon^{O(\lambda)}$, where $\varepsilon$ is the parameter in the $(1+\varepsilon)$-approximation and $\lambda$ is the doubling dimension of the dataset. On the practical side, the paper used a host of heuristic methods inspired by their theoretical result and obtained state-of-the-art empirical performances.

The main techniques of the paper are quite straightforward: it combines the subset-query algorithm of CIP [ICALP’02] with LSH and graph-based nearest-neighbor search algorithms. It’s interesting to see that some decade-old results in algorithm design could be helpful for modern machine learning applications.

**Strengths:**

In general, I have a positive opinion of this paper: the problem it studied is extremely relevant to modern machine learning applications, the paper is well-written, and it contains a very cute connection between theory and practice. The ‘lessons’ listed in the paper are quite clear and should be easy to digest for practitioners (in my opinion, although I work on the more theoretical side), and the theory-inspired practical approach outperformed state-of-the-art empirical algorithms. I enjoyed a good read of the paper, and I would advocate for its acceptance.

**Weaknesses:**

- Some writing issues of the focus of the paper. The paper contains many results; the title and abstract suggest that the graph-based approach is the main contribution of the paper. However, looking at section 4, it appears that on the theoretical front, the key idea is to adapt the algorithm of CIP [ICALP’02] instead. This is also emphasized in ‘lesson 2’ which appears to say that query planning as in CIP [ICALP’02] is equally important as graph-based approaches. I understand summarizing the key points is often hard when the paper contains many results; however, I think the current story is still a bit unfocused.
- The $o(2^m)$ memory bound is somehow weak: essentially, the result is only $o(2^m)$ if $\delta\leq 1/\log^{2}{m}$, and it’s unclear what is the leading constant there. As such, I’m not really convinced that this is the right way to go in theory.
- One potential criticism could be that the techniques used in this work are quite straightforward and are essentially a combination of existing results. I think it is quite fine for ML conferences like ICLR; however, the authors would be able to address this if other readers flag it as a criticism.

**Questions:**

- Do you have comments about the hidden constant in the exponent of $2^{O(m\sqrt{\delta}\log^{2}{m})}$?
- Also, in Theorem 4.2, the bound would hold only for $\varepsilon<1$ right? Say that if I only want a $10$ approximation, how would the bound scale?
- For the definition of the doubling dimension of datasets, I would suggest having more discussions after the definition to help the readers understand the notion better (this is a MISC comment).
- The problem studied in this paper emphasizes that the labels of the required query vector $q$ have to be a subset of the given set $S_q$. If we instead only want $S_v \cap S_q \neq \emptyset$, or something like $|S_v \cap S_q|\geq  0.1\cdot |S_q|$, does the problem changes drastically?

---

### Official Review · Reviewer_Qt5Y · 2024-11-03

**Soundness:** 2
**Presentation:** 2
**Contribution:** 2
**Rating:** 3
**Confidence:** 4

**Summary:**

The paper studies a constrained variant of approximate nearest neighbor search where the labels of a query vector must be a subset of the labels of each retrieved vector. The authors prove theoretical bounds for the problem based on both earlier work on the subset query problem and earlier bounds for LSH and graph-based methods. The authors also propose an improved empirical algorithm based on FilteredDiskANN for the problem.

**Strengths:**

- The paper studies an important practical problem encountered in many real-world use cases of nearest neighbor search. Currently, the problem of incorporating multiple filters into the standard vector search problem is not addressed well by any of the existing methods.
- The paper makes fairly interesting novel contributions to the problem both in theory and in practice. In particular, the connection to the subset query problem and the analysis using the CIP data structure is quite interesting and the proposed penalty search method seems to work well in practice.

**Weaknesses:**

Instead of studying the performance of a practical multiple filter nearest neighbor search algorithm, the authors design a theoretical algorithm using (what the authors refer to as) the CIP data structure for the subset query problem but with the final groups in the structure replaced with ANN indexes. However, while connecting these is quite neat, the theory is not really insightful and I don't see how these results inspire the empirical algorithm presented in Section 6.

The authors write that there are two lessons to be learned from the theoretical analysis, the first one being that graph-based ANN algorithms are powerful for the multi-filter variant. However, just because you can get theoretical guarantees by using them in the last level of the CIP data structure (as you can with any ANN index), does not logically imply that purely graph-based methods would be specifically suited for the problem. The underlying hypothesis is actually simply that since graph-based methods are SOTA for standard ANN, they probably are also for the multi-filter variant. The second given lesson is that query planning is important. However, routing queries to relevant subsets is a natural part of any ANN algorithm and I do not see any specific connection between the routing in CIP and your proposed heuristic query planning method. It would be helpful for the authors to clarify any possible connections beyond these lessons.

Instead, the paper proposes a heuristic method building upon an existing method (FilteredDiskANN) by introducing the penalty search method and query planning to the algorithm. The penalty search is quite interesting, but in practice has to be approximated and refined in an ad hoc way, introducing two additional hyperparameters, $\lambda$ and $\tau$. The proposed query planning method is also very ad hoc and not studied well in detail (see below). The authors also provide instances where prior algorithms fail. In some cases studying such failure cases leads to further insight, but I do not find these particular cases to be interesting from the theory point of view and in practice the nature of ANN search is that we mostly do not really care about the worst case as long as the methods work on real-world datasets (for which many algorithms work well regardless of e.g. the curse of dimensionality).

For the experiments, the authors should be more clear as to why they do not compare against FilteredDiskANN (it does not work well for more than one filter?). There is an ablation on the effect of the penalty search on one data set, but you would need to compare against FilteredDiskANN on all data sets, and you also need to have an ablation study on the effect of the query planning. You can apply the query planning to any of the algorithms; right now I cannot tell which part of your method affects its performance the most. In contrast, I do not find the ablation studies for the query correlation or the query filter selectivity to be that interesting. Finally, you should analyze more deeply why your method works better than the other methods on the dataset you introduce as opposed to e.g. the Amazon data set where there is no improvement compared to ACORN. The provided new data set is quite interesting but in the abstract, the authors claim to introduce multiple novel datasets (I would not count the synthetic datasets here).

The paper is quite hard to read with multiple issues in the presentation. To improve the presentation, the authors should at least
  - Refer to the relevant Appendix where proofs can be found, e.g. for Lemmas 5.1-5.3.
  - Add "for every $\delta > 0$" to the statement of Theorem 4.1
  - Use \citep for citations, currently the lack of parentheses makes the text difficult to read
  - Explain properly how prior work relates to the paper instead of just giving a long list of citations (Section 3)
  - Give a name for your algorithm such that it is easier to refer to
  - Make sure figures that are next to each other are of the same size
  - Include a discussion section

**Questions:**

- Is your theory only applicable to $\ell_2$ as the proof of Theorem 4.1 would imply?
- How do you select the cut-offs for your proposed query planning method? Are they the same for each dataset?
- How difficult are the hyperparameters $\lambda$ and $\tau$ to set?
- Which data set is used for the ablation study on removing the penalty search?
- Why does ParlayIVF2 only have one instance on the Pareto frontier in two of the experiments?

---

> ### Author Response · Authors · 2024-11-26
>
> We thank the reviewer for their thoughtful comments regarding the weaknesses of our submission. All presentation issues will be corrected in a final version.
>
> Below, we try to give a high-level explanation of our theoretical framework, and its relevance to our empirically supported algorithms.
> To put it concisely, our theoretical contributions are two-fold: first, we give the first mathematical formalization of the practically important MultiFilterANN problem by connecting it with subset query and the classical ANNS problems. Second, and more importantly, our results show that the difficulty of the MultifilterANN problem is mostly dominated by the subset query requirement. This is why the stated bounds of Theorem 4.2 generalize Theorem 4.1. Indeed, the subset query problem (i.e. given n base subsets of some universe U, and a query subset Sq, find at least one base subset Sv which is a superset of Sq) is very challenging from a theoretical perspective, and Theorem 4.1 is the state-of-art for the problem. In the proof of Theorem 4.2, we show that it is possible to combine a good ANN structure cleverly with the CIP-based partitioning data-structure for subset query to solve MultifilterANN.
>
> But how does this translate to real-world algorithms?
>
> We conduct a rigorous undertaking by studying two plausible candidate algorithms for MultifilterANN, namely ParlayIVF2 (a clustering-based algorithm for MultifilterANN) and Filtered-Vamana (a graph-based algo that apparently only works for queries coming with single filters as opposed to AND predicates). We demonstrate different bad examples for both in worst-case guarantees. These examples motivate our final algorithm, which a) combines clustering and graph-based methods to avoid these bad examples, and b) modifies the graph search algorithm (which has invariably been a greedy algorithm in prior work) to escape the bad examples we constructed.
> Finally, we note that prior work on the 1 filter case, such as FilteredDiskANN, is not directly applicable to the multi-filter setting. While it is true that we can reduce the multi-filter problem to the one filter setting, this incurs an exponential blowup in the number of labels, which is prohibitive and impractical for all of our datasets.

---

> ### Author Response · Authors · 2024-11-26
>
> We also thank the reviewer for the raised questions. We address them below:
>
> > - Is your theory only applicable to ℓ2 as the proof of Theorem 4.1 would imply?
>
> No, Theorem 4.1 also holds for many other distances that have efficient underlying nearest neighbor data structures, such as $\ell_1$ and $\ell_p$ norms between $1$ and $2$. However the most practically motivated setting is $\ell_2$.
>
> > - How do you select the cut-offs for your proposed query planning method? Are they the same for each dataset?
>
> The cutoffs are based on the data dimension and the selectivity of the query filter's predicate (i.e. how often the predicate appears in the document data). In general, around 30,000 vectors in 100 dimensions can be brute-forced in a few milliseconds, and so we scale it depending on the dimension (e.g., 10000 for 300 dims). For the clustering threshold, we use a slightly larger number (50000) to ensure that only the queries with commonly occurring predicates get routed to the graph algorithm.
>
> > - How difficult are the hyperparameters $\lambda$ and $\tau$ to set?
>
> We have used only one set of fixed values in the paper – $\lambda$ depends on the scale of distances, and we set it to 10, since the norms were all close to the unit scale. The value of $\tau$ was fixed to 1 in this paper, to allow one filter mismatch. This we suspect depends on the number of clauses in the query predicate. Our algorithm is quite robust to $\lambda$ value, and we could not test other $\tau$ values due to lack of datasets with multiple query clauses. We will add some ablation studies by varying these parameters which will indicate how robust they are.
>
> > - Which data set is used for the ablation study on removing the penalty search?
>
> We used the Wikipedia dataset, and will state it clearly in a final submission. We thank the reviewer for pointing this out.
>
> > - Why does ParlayIVF2 only have one instance on the Pareto frontier in two of the experiments?
>
> We ran a large parameter sweep for ParlayIVF2 and ACORN, and in these cases described by the reviewer, their code simply resulted in a cluster of points all in the same latency-recall regime, so we simply reported one of these numbers.

---

> > ### Comment · Reviewer_Qt5Y · 2024-11-28
> >
> > Thank you for your answers to my questions. However, your reply does not really answer the concerns I raised in my review. In particular, the connection between the theory presented in Section 4 and your empirical algorithm is tenuous. Section 5 is more motivating, but in the end your empirical algorithm is still a very ad hoc combination of brute-force search, clustering, and graph search. Your experimental section would need to have a proper ablation study on the effect of each of these components as now no one can tell which component gives your method the edge over the other methods (I suspect it is the brute force search component).

---

### Official Review · Reviewer_UJzq · 2024-11-04

**Soundness:** 2
**Presentation:** 2
**Contribution:** 2
**Rating:** 3
**Confidence:** 4

**Summary:**

The article considers a task the authors call MultiFilterANN. MultiFilterANN extends standard approximate nearest neighbor (ANN) search by assuming that both the query point and the database points have labels from the discrete label space $[m]$, and setting a hard constraint that the label set of the query point must be a subset of any (approximate) nearest neighbor returned by the algorithm. The article proposes an algorithm that extends earlier work on graph-based filtered ANN search. Specifically, they propose searching a FilteredDiskANN graph (Gollabudi et al. 2023) by greedy penalty search (Wang et al., 2024), where the symmetric label distance used in the penalty computations is replaced by the asymmetric label distance. According to the empirical evaluation, the proposed method outperforms the earlier filtered ANN methods ParlayIVF$^2$ and ACORN in the special case of two filters. The article also discusses the limitations of earlier filtered ANN methods in the case of multiple filters, and provides toy examples where these methods perform badly. In addition, theoretical guarantees are presented for a different MultiFilterANN algorithm (that combines a graph or LSH index with a CIP data structure for a subset query problem).

Gollapudi, Siddharth, et al. "Filtered-diskann: Graph algorithms for approximate nearest neighbor search with filters." Proceedings of the ACM Web Conference, p. 3406-3416. 2023.

Wang, Mengzhao, et al. "An efficient and robust framework for approximate nearest neighbor search with attribute constraint." Advances in Neural Information Processing Systems 36 (2024).

**Strengths:**

The idea of replacing the symmetric label distance with the asymmetric label distance on the greedy search of NHQ algorithm (Wang et al., 2024) is intuitive, and it self-evidently improves performance when the label constraint are subset queries ($S_q \subset S_v$) instead of exact matches ($S_q = S_v$).

Wang, Mengzhao, et al. "An efficient and robust framework for approximate nearest neighbor search with attribute constraint." Advances in Neural Information Processing Systems 36 (2024).

**Weaknesses:**

The authors propose (in Section 6) a graph-based algorithm for MultiFilterANN problem (I explain my interpretation of the proposed method in Summary). However, it seems to this reviewer that the proposed algorithm is supported neither by (a) theoretical results, nor by (b) empirical evaluation. The theoretical results are for the different algorithm (described in Section 4) and it may be that the algorithm that is evaluated empirically in Section 7 is a yet different algorithm (described in Appendix E).

Specifically, the article starts by reviewing a CIP data structure (Charikar et al., 2002) for subset query problem and then proposing a MultiFilterANN algorithm that combines it with either a graph or LSH index structure. However, later in Section 6 it turns out that the authors propose a completely different graph-based empirical algorithm that does not use a CIP data structure at all. Thus it seems that theoretical analysis of Section 4 is disconnected from the rest of the article (except that MultiFilterANN problem is considered), and does not offer theoretical support for the proposed method. The authors should clarify what is the connection between the empirical algorithm and the theoretical discussion of Section 4.

Another part that must be clarified significantly is the empirical evaluation in Section 7. In particular, it is unclear whether the results are obtained by using the graph-based method proposed in Section 6 or the hybrid methodology described in Appendix E? According Appendix E, when the estimated number database points satisfying the constraints is small, either a brute force search after identification of these points, or a clustering-based approach is used instead of the graph-based method proposed in Section 6. If the hybrid methodology of Appendix E is used to generate the results of Section 7, this does not yet tell anything about the performance of the proposed graph method, since it is possible that the results are obtained using brute force search or a clustering-based approach.

You should include a direct comparison between the proposed graph method and the earlier filtered ANN methods (ParlayIVF$^2$ and ACORN) (assuming the results of Section 7 are for the hybrid method described in Appendix E). In particular, when the number of database points satisfying the constraints is small, I suspect that the brute force baseline is the fastest method, and if you enable this option for your method, but not for ParlayIVF$^2$ and ACORN, this distorts the results. In addition, you should include ablation experiments documenting in which regime your method outperforms this simple baseline (that has an additional benefit of producing exact results).

It is also not clear how relevant the studied problem (MultiFilterANN) is. I am not deeply familiar with literature on filtered ANN search, but it seems that the recent related works cited by the authors only consider the special cases of one or two filters. In particular, I suspect that when there are multiple filters (for instance, $|S_q| \geq 3$), the simplest possible ad hoc approach (brute force search after finding the database points with correct labels) would often outperform more complicated filtered ANN methods, since in this case the set of database points matching all of query labels is probably quite small.

Presentation is low quality. The article does not seem carefully finished. For instance, citations should be inside the parenthesis when the author names are not used as a part of the text. For instance, you should write "we may use Locality Sensitive Hashing (Indyk & Motwani 1998a)" instead of "we may use Locality Sensitive Hashing Indyk & Motwani 1998a". Just use citep-command if you are using natbib. As an another example, there is a list of 26 (!) consecutive citations in the beginning of Related work-section. It seems that the only relevance of these works is that they are connected to ANN search. You should be much more specific in Related work-section and do not use this kind of generic lists.

Charikar, M., Indyk, P., & Panigrahy, R. (2002, June). New algorithms for subset query, partial match, orthogonal range searching, and related problems. In International Colloquium on Automata, Languages, and Programming (pp. 451-462). Berlin, Heidelberg: Springer Berlin Heidelberg.

**Questions:**

Can clarify how the theoretical results of Section 4 are connected to the empirical algorithm you propose in Section 6? It seems to this reviewer that the algorithm considered in Section 4 (2-stage search where you first prune a candidate set on the label space $[m]$ with a CIP data structure, and then perform a nn-search on the continuous feature space $\mathbb{R}^d$ with either LSH or a graph) is completely different than the one you propose in Section 6 (FilteredDiskANN graph from earlier work (Gollabudi et al., 2023) combined with greedy search penalized by asymmetric label distance)?

Are the results of Section 7 for the graph-based approach of Section 6 or for the hybrid approach of Appendix E that also can use either brute force search or clustering?

Gollapudi, Siddharth, et al. "Filtered-diskann: Graph algorithms for approximate nearest neighbor search with filters." Proceedings of the ACM Web Conference, p. 3406-3416. 2023.

---

> ### Author Response · Authors · 2024-11-26
>
> We thank the reviewer for their thoughtful comments and concerns. Below, we address their concerns (split over multiple comments).
>
> > The authors propose (in Section 6) a graph-based algorithm for MultiFilterANN problem ... The authors should clarify what is the connection between the empirical algorithm and the theoretical discussion of Section 4.
>
> We believe a major contribution of our theoretical results is conceptual: we connect the widely studied nearest neighbor problem to the theoretically challenging problem of subset query, to give the first formalization of the practically important MultiFilterANN problem. A priori, it is not clear how these two separate components (nearest neighbor search and subset query) are connected or interact. Conceptually, our theoretical results show that the difficulty of the problem is dominated by the subset query requirement.
> This is not an artifact of theory: in our experiments we also observe that as the subset query constraints become more involved, the performance of all methods degrade. Furthermore, the lessons learned from our theoretical algorithm (lessons 1 - 4) directly translate into our empirical algorithm design process, e.g., Theorem 4.1 and Section 4 show that some form of query planning is needed which is a lesson we use in our empirical algorithm design. Furthermore, Theorem 6.1, and Lemma D.1 gives provable guarantees about the specific graph algorithm we use in our experiments. The theorem shows how we can bound the doubling dimension of such a new metric using the original dimension, and DiskANN is the only known graph algorithm which comes with provable guarantees.

---

> ### Author Response · Authors · 2024-11-26
>
> > Another part that must be clarified significantly is the empirical evaluation in Section 7. In particular, it is unclear whether the results are obtained by using the graph-based method proposed in Section 6 or the hybrid methodology described in Appendix E?
>
> We thank the reviewer for the comment. For the experiments, we do use the hybrid approach described in Appendix E, which is well-motivated by lessons 2 (query planning) and 3 (graph index benefits) from sections 4 and 5.
> Our experiments are comprehensive and demonstrate the need for all the approaches considered. For example, in The Wikipedia Dataset (Fig 1, left) and YFCC dataset (Fig 1, right), our hybrid algorithm and ParlayIVF perform the best, perhaps because of the need for bruteforce and clustering methods due to the filter specificity being less. On the Amazon Dataset, our approach and ACORN does the best. In terms of baselines, ParlayIVF is a bruteforce+clustering approach, while ACORN is a purely graph-based method.
> That said, you are correct that as written, the results could entirely be obtained using brute force or clustering-based searches, so we will add a table indicating how the queries are split across the three methods used in the hybrid search. Figure 6 illustrates how our purely-graph algorithm performs as a function of query specificity, with dense predicates (having many base points satisfying the predicate) performing best, and the rare filters are not as great. We have made a nomenclature error with the interpretation of specificity in the figure though, and will fix it. The best-performing curve is for queries whose predicates are satisfied by a large fraction of base points.

---

> ### Author Response · Authors · 2024-11-26
>
> > You should include a direct comparison between the proposed graph method and the earlier filtered ANN methods (ParlayIVF and ACORN) (assuming the results of Section 7 are for the hybrid method described in Appendix E).
>
> We thank the reviewer for the comment. ParlayIVF indeed is based on query planning, and has brute force or clustering to route to. Their approach does not use graphs for multiple filters, and resorts to graph-indices only for single filters at query time. As for ACORN, a scan of the code indicates that it does not include a brute-force layer, and relies only on graph indices. That said, the ACORN codebase assumes that the complete filter map of the set of points which satisfy the query predicate is known up front, and that is not counted in their reported latency numbers. While this operation is small at small scales, we suspect that this computation would quickly add to, and even dominate,  the overall query latency for large datasets and complex predicates. Our approach, however, is much more forgiving to this filter-evaluation complexity, as it is only done on the fly, as the graph is being searched. We take your point though, and will try to include agnostic comparisons like number of distance comparisons vs recall, between ACORN graph and our graph algorithm in the final version of the paper, to obtain a more apples-to-apples comparison.

---

> ### Author Response · Authors · 2024-11-26
>
> > It is also not clear how relevant the studied problem (MultiFilterANN) is. I am not deeply familiar with literature on filtered ANN search, but it seems that the recent related works cited by the authors only consider the special cases of one or two filters. In particular, I suspect that when there are multiple filters (for instance, |Sq|≥3), the simplest possible ad hoc approach (brute force search after finding the database points with correct labels) would often outperform more complicated filtered ANN methods, since in this case the set of database points matching all of query labels is probably quite small.
>
> We thank the reviewer for raising a fair and reasonable concern. For MultifilterANN, the filter predicates that come with queries also come with a specificity, which denotes how common the valid documents are for that predicate within the document corpus. For small-scale demonstration experiments in research, the document corpus rarely exceeds a size of 10-50 million, and for |S_q| \geq 3, it would not be uncommon to expect 100-200 valid documents, for which a simple brute-force search is more than enough. However, in practice, the document corpus can easily reach a size in the magnitude of billions; here, valid documents for a sufficiently large predicate would now likely range in the millions. This is the primary use-case for a graph-based MFANNS approach (see FilteredDiskANN and ACORN), and we believe that our overall approach would shine here. The problem is very relevant [1, 2, 3, 4], and almost all real-world use-cases of ANN now come with additional filtering, and all the vector database services are forced to add this feature.
>
> An analogous claim about scale has been made for multiple variants of the broader ANNS problem: for example, DiskANN, FilteredDiskANN, FreshDiskANN, and OOD-DiskANN [5, 6, 7, 8] each have sections devoted to performance on large-scale datasets. However, there is not enough publicly available data to support a similar claim for the multifilter setting – this remains a major open problem within the space in order to further research. With that said, this warrants additional discussion, and we can include it in a final version of our submission.
>
> 1. https://www.pinecone.io/learn/vector-search-filtering/
> 2. https://weaviate.io/developers/weaviate/search/hybrid
> 3. https://qdrant.tech/articles/hybrid-search/
> 4. https://docs.trychroma.com/guides#filtering-by-document-contents
> 5. https://papers.nips.cc/paper_files/paper/2019/file/09853c7fb1d3f8ee67a61b6bf4a7f8e6-Paper.pdf
> 6. https://harsha-simhadri.org/pubs/Filtered-DiskANN23.pdf
> 7. https://arxiv.org/pdf/2105.09613
> 8. https://arxiv.org/pdf/2211.12850

---

### Note · Authors · 2024-12-03

**Comment:**

Thank you to all the reviewers for their valuable feedback.

**Withdrawal Confirmation:**

I have read and agree with the venue's withdrawal policy on behalf of myself and my co-authors.